# Depressed 660-km discontinuity caused by akimotoite–bridgmanite transition

Artem Chanyshev[1,2 ✉], Takayuki Ishii[2,3 ✉], Dmitry Bondar[2], Shrikant Bhat[1], Eun Jeong Kim[2], Robert Farla[1], Keisuke Nishida[2], Zhaodong Liu[2,4], Lin Wang[2,5], Ayano Nakajima[6], Bingmin Yan[3], Hu Tang[3], Zhen Chen[3], Yuji Higo[7], Yoshinori Tange[7] & Tomoo Katsura[2,3]

The 660-kilometre seismic discontinuity is the boundary between the Earth's lower mantle and transition zone and is commonly interpreted as being due to the dissociation of ringwoodite to bridgmanite plus ferropericlase (post-spinel transition)[1–3]. A distinct feature of the 660-kilometre discontinuity is its depression to 750 kilometres beneath subduction zones[4–10]. However, in situ X-ray diffraction studies using multi-anvil techniques have demonstrated negative but gentle Clapeyron slopes (that is, the ratio between pressure and temperature changes) of the post-spinel transition that do not allow a significant depression[11–13]. On the other hand, conventional high-pressure experiments face difficulties in accurate phase identification due to inevitable pressure changes during heating and the persistent presence of metastable phases[1,3]. Here we determine the post-spinel and akimotoite–bridgmanite transition boundaries by multi-anvil experiments using in situ X-ray diffraction, with the boundaries strictly based on the definition of phase equilibrium. The post-spinel boundary has almost no temperature dependence, whereas the akimotoite–bridgmanite transition has a very steep negative boundary slope at temperatures lower than ambient mantle geotherms. The large depressions of the 660-kilometre discontinuity in cold subduction zones are thus interpreted as the akimotoite–bridgmanite transition. The steep negative boundary of the akimotoite–bridgmanite transition will cause slab stagnation (a stalling of the slab's descent) due to significant upward buoyancy[14,15].

The 660-km discontinuity (D660) is one of the most important structural boundaries in the Earth's mantle, which plays an essential role in mantle dynamics because it determines whether slabs stagnate or penetrate into the lower mantle[14]. Seismological observations reveal significant depressions of the D660 down to 750 km, as well as a D660 multiple-discontinuity structure in subduction zones[4–10]. Such observations should be related to the phase transitions of mantle minerals. Because slabs are generally colder than the ambient mantle, the temperature dependence of phase transitions (the Clapeyron slope in the pressure–temperature ($P$–$T$) phase diagram) is essential to interpret such depressions and understand subduction dynamics.

The D660 is usually attributed to the dissociation of $(Mg,Fe)_2SiO_4$ ringwoodite (Rw) into $(Mg,Fe)SiO_3$ bridgmanite (Brg) and $(Mg,Fe)O$ ferropericlase (fPc) (the RBP transition, hereafter)[1–3]. Recent in situ X-ray diffraction studies using a multi-anvil apparatus showed that this reaction has negative but gentle Clapeyron slopes ($-1.3$ MPa K$^{-1}$ to $-0.5$ MPa K$^{-1}$), which can only vary the D660 depth between 630 and 670 km depth[11–13,16]. An alternative explanation of the D660 depressions in subduction zones is therefore needed. Currently, the following three hypotheses are being discussed. One is steepening of the RBP phase boundary by water. One study[12] suggested that the

addition of 2 wt% of water produces a phase boundary with a slope of $-3.1$ MPa K$^{-1}$ to $-3.2$ MPa K$^{-1}$. However, this suggestion could be invalid because of improper experimental procedure; the formation of ringwoodite from bridgmanite plus ferropericlase was not demonstrated under wet conditions in this study. The second explanation is the persistence of metastable Rw in the Brg + fPc stability field owing to sluggish kinetics at low temperatures[17,18]. One seismological study demonstrated that the D660 is extremely sharp and less than 2 km thick in cold subduction zones[9], and such sharpness can be interpreted as being due to the overpressure-induced RBP transition only at extremely low temperatures ($<1,000$ K)[18]. Although this explanation is possible, there has been no determination of the transition kinetics at such low temperatures to provide a definite answer. The third is that the depressed D660 is caused by another phase transition with a steeper Clapeyron slope. The most likely candidate is the akimotoite (Ak)–Brg transition (AB transition, hereafter), as previously proposed[19]. High-pressure and high-temperature in situ X-ray studies have shown that this transition occurs at a similar pressure to the RBP transition, but has a steeper slope ($-3.2$ MPa K$^{-1}$)[20–22]. Hence, the AB transition could be responsible for the D660 depressions beneath subduction zones.

[1]Deutsches Elektronen-Synchrotron DESY, Hamburg, Germany. [2]Bayerisches Geoinstitut, University of Bayreuth, Bayreuth, Germany. [3]Center for High Pressure Science and Technology Advanced Research, Beijing, China. [4]State Key Laboratory of Superhard Materials, Jilin University, Changchun, China. [5]Earth and Planets Laboratory, Carnegie Institution, Washington, DC, USA. [6]Department of Earth Sciences, Graduate School of Science, Tohoku University, Sendai, Japan. [7]Japan Synchrotron Radiation Research Institute (JASRI), Sayo, Japan. ✉e-mail: artem.chanyshev@uni-bayreuth.de; takayuki.ishii@hpstar.ac.cn

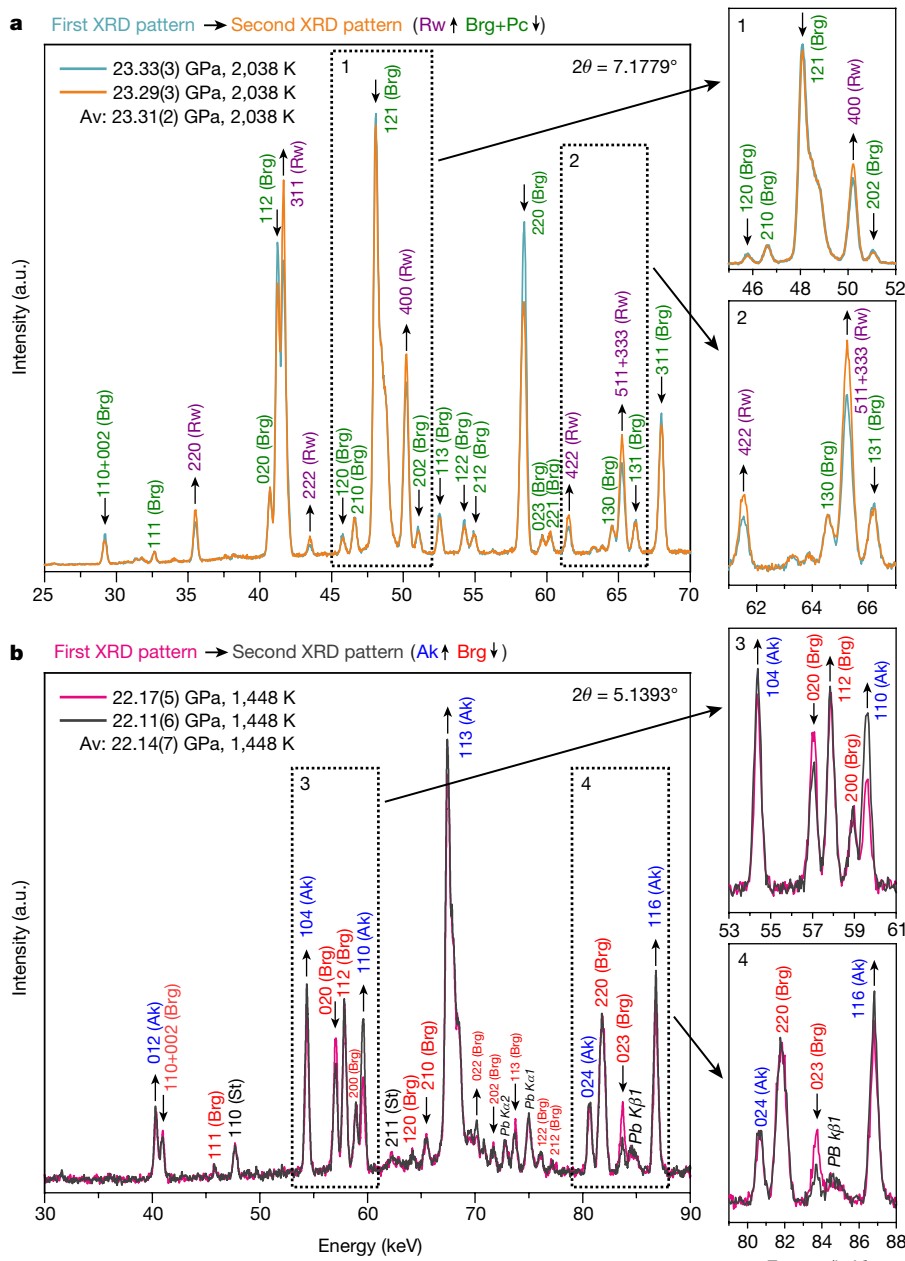

**Fig. 1 | Accurate phase identification by means of in situ X-ray diffraction in a multi-anvil press. a**, An example of the change of intensity ratio between ringwoodite and bridgmanite plus periclase at 2,038 K and 23.31(2) GPa. **b**, An example of the change of intensity ratio between akimotoite and bridgmanite at 1,448 K and 22.14(7) GPa. The first diffraction patterns are shown in (**a**) blue and (**b**) pink, and the second patterns are displayed in (**a**) orange and (**b**) dark grey. Magnified areas of the diffraction patterns are shown on the right. The numbers above the peaks indicate the Miller indexes of bridgmanite, ringwoodite, akimotoite and stishovite. The upward and downward arrows indicate peaks with increased and decreased intensities in the second diffraction patterns, respectively. Because the intensities of most peaks of the low-pressure phases increase in the second diffraction patterns, the stable phases were defined as (**a**) ringwoodite at 23.29(3) GPa and 2,038 K, and (**b**) akimotoite at 22.11(6) GPa and 1,448 K. In **b**, the fluorescence lines of Pb Kα and Kβ are shown by the Siegbahn notation. Av., average.

Combining the multi-anvil technique with in situ X-ray diffraction produces the most reliable high $P$–$T$ phase relations data because of the well-controlled $P$–$T$ conditions[23]. Nevertheless, most previous experiments using this combination investigated phase stability based only on the formation of new phases from a starting material that is stable under ambient conditions. Such data indicate that the newly formed phases are more stable than the starting material but do not provide any information about phase equilibrium between high-pressure phases. The definition of phase equilibrium is the balance between

the forward reaction (lower-pressure phase to higher-pressure phase) and reverse reaction (higher- to lower-pressure phase). The stability of the lower- and higher-pressure phases must therefore be determined by bracketing based on the results of a pair of forward and reverse reactions. Furthermore, the kinetics of phase transitions of mantle minerals becomes extremely sluggish once a high-pressure phase is formed[13,22]. Our recent studies[1,3] indicated that conventional high $P$–$T$ experiments could have misinterpreted phase relations owing to sudden and large changes in sample pressure upon heating and a

sluggish transformation of a newly formed phase. It is thus necessary to re-investigate the previously determined phase relations using a strategy that considers the above issues.

In this study, we determined the boundaries of the RBP and AB phase transitions in the MgO–SiO$_2$ systems over a temperature range of 1,250–2,085 K by using advanced multi-anvil techniques with in situ X-ray diffraction. The detailed experimental procedure is described in the Methods. The crucial point of our strategy is that the phase stability was determined from the relative change of the diffraction peak intensities of the coexisting lower-pressure and higher-pressure phases at a fixed press load and temperature to observe the growth of a stable phase. This method enables us to correctly determine the reaction direction by removing the kinetic effects and to avoid sudden pressure changes upon heating or cooling[3] (Fig. 1). Although the determination of phase stability by comparing in situ X-ray diffraction intensities at a fixed temperatures was previously done[16], that study lacked data on the paired forward and reverse reactions at the same temperatures and did not consider the effect of kinetics.

The RBP boundary has a slightly concave curve, whereas the AB boundary has a steep convex curve (Fig. 2). The RBP boundary is located at pressures of 23.2–23.7 GPa in the temperature range of 1,250–2,040 K. Its slope varies from −0.1 MPa K$^{-1}$ at temperatures less than 1,700 K to −0.9 MPa K$^{-1}$ at 2,000 K with an averaged value of −0.5 MPa K$^{-1}$ (Fig. 2). The slope of the AB boundary gradually changes from −8.1 MPa K$^{-1}$ at low temperatures up to 1,300 K to −3.2 MPa K$^{-1}$ above 1,600 K. The AB boundary is located at higher pressure than the RBP boundary at $T$ < 1,260 K (based on linear extrapolation), where $P$ = 23.8 GPa (Fig. 3). In this temperature range, Rw should dissociate not into Brg but Ak plus periclase (Pc). The AB boundary will be located at $P$ = 26.8 GPa and $T$ = 900 K by linear extrapolation of our data below 1,350 K.

The high precision of pressure and temperature determination, high density of experimental data points and application of the advanced techniques to avoid kinetic problems allowed us to determine phase boundaries far more accurately than in previous studies. The majority of previous studies after 2003 on the RBP boundary yielded similar slopes but a lower-pressure location than the present study (Extended Data Fig. 1a). The lower-pressure boundaries[12,13,16] were obtained using gold pressure scales, whereas the previous study that used a MgO scale[11], which is also adopted in this study, was located at similar pressure. The quench (ex situ) experiments[2,21] exhibited steep slopes and higher pressures. The previous experimentally determined AB boundaries[2,20,22,24,25] yielded similar slopes of the high-temperature side of the present boundary (Extended Data Fig. 1b), whereas the steep slope in the low-temperature side has not been investigated previously. Earlier experimental studies were not able to overcome the sluggish kinetics of the AB transition at low temperature and obtain the correct phase boundary, as explained above. Theoretical studies predicted steeper slopes of the AB boundary, −3.5 ± 0.8 MPa K$^{-1}$ (ref. [26]) and −6.0 ± 1.0 MPa K$^{-1}$ (ref. [27]), than those determined experimentally. Thus, our results are in better agreement with theoretical data than are previous experimental studies.

## Curvature of the AB transition boundary

We find that the Clapeyron slope of the AB phase transition boundary gradually varies with temperature. The slope of a phase boundary is equal to the ratio of the entropy change ($\Delta S_{\mathrm{tr}}$) to the volume change ($\Delta V_{\mathrm{tr}}$) associated with the phase transition according to the Clausius–Clapeyron relation: $\mathrm{d}P/\mathrm{d}T = \Delta S_{\mathrm{tr}}/\Delta V_{\mathrm{tr}}$. The volume changes of the AB transition vary only slightly over the entire investigated temperature range because both phases have similar thermal expansion coefficients and bulk moduli (Extended Data Table 1). The entropy is a function of isobaric heat capacity ($C_P$) and temperature $S_T^0 = S_{298}^0 + \int_{T_0}^{T} \frac{C_P}{T}\mathrm{d}T$. Calorimetric studies have reported Ak and Brg heat capacities only up to 700 K (ref. [28]) and 295 K (ref. [29]), respectively, owing to the limited

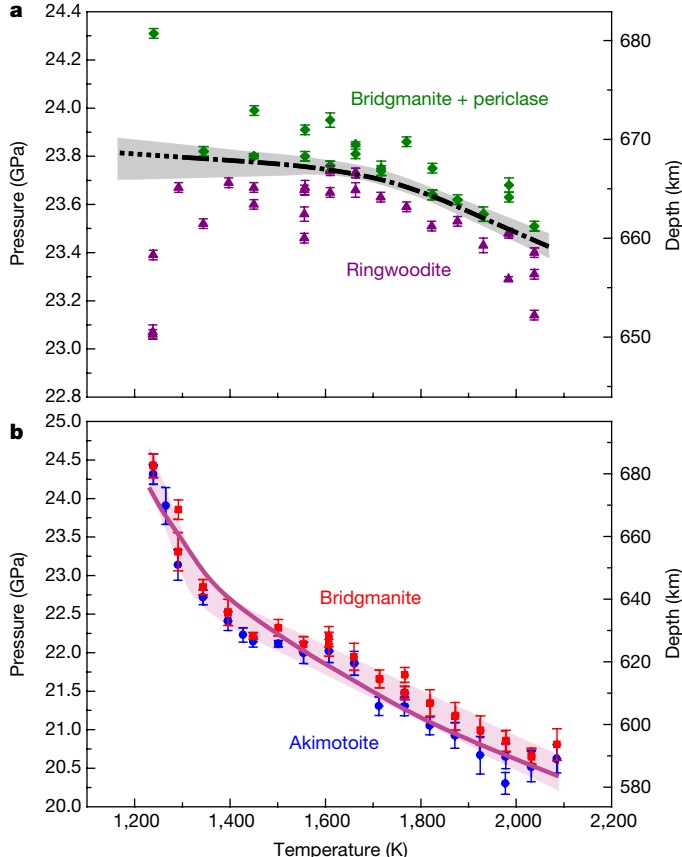

**Fig. 2 | Phase relations of bridgmanite-forming transitions in MgO-SiO$_2$ systems. a**, RBP transition in the Mg$_2$SiO$_4$ system. Green diamonds and purple triangles indicate the $P$–$T$ conditions at which the ratio of bridgmanite plus periclase to ringwoodite increases or decreases, respectively. The grey shaded area indicates an allowed region for the RBP boundary. The dot-dashed curve represents the most probable transition curve within this area. **b**, AB transition in the MgSiO$_3$ system. Red squares and blue circles indicate the $P$–$T$ conditions at which the ratio of bridgmanite to akimotoite increases or decreases, respectively. The violet shaded area indicates an allowed region for the AB boundary for the vast majority of experimental data points. The solid violet curve represents the most probable transition curve within this area. The probable reason for variation of the AB phase transition boundary is given in the text. Pressures were determined from the MgO unit-cell volumes using the Birch–Murnaghan and Vinet equations of state from ref. [47]. Error bars originate from the pressure uncertainty of the MgO equation of state suggested from ref. [47]. The temperatures and pressures were corrected based on the pressure effects on the thermoelectromotive force of a W$_{97}$Re$_3$–W$_{75}$Re$_{25}$ thermocouple[48].

thermal stability of these minerals at ambient pressure. A few theoretical studies calculated their heat capacities[26,30], but according to these calculations the AB boundary should have a relatively constant Clapeyron slope of −3.5 ± 0.8 MPa K$^{-1}$ (ref. [26]), which contradicts our results. Thus, further explanation is required. The curved AB boundary can be explained by different curvatures of Ak and Brg heat capacities with temperature (Extended Data Fig. 2), as suggested by one computational study[30]. We estimate that the Ak heat capacity should be lower than that of Brg at $T$ < 860 K and higher at higher temperatures (Extended Data Fig. 2). This difference in curvature could result from temperature-dependent cation disorder in Ak. Although possible cation disorder in Ak has been previously suggested[31–33], no X-ray diffraction measurement has experimentally shown the Mg–Si disorder population[34]. Nevertheless, the measured distances of Mg–O (2.077 Å; ref. [34]) and Si–O (1.799 Å; ref. [34]) in Ak and Shannon's ionic radii of Mg (0.72 Å; ref. [35]), Si (0.4 Å; ref. [35]) and O (1.4 Å; ref. [35]) imply that Si may

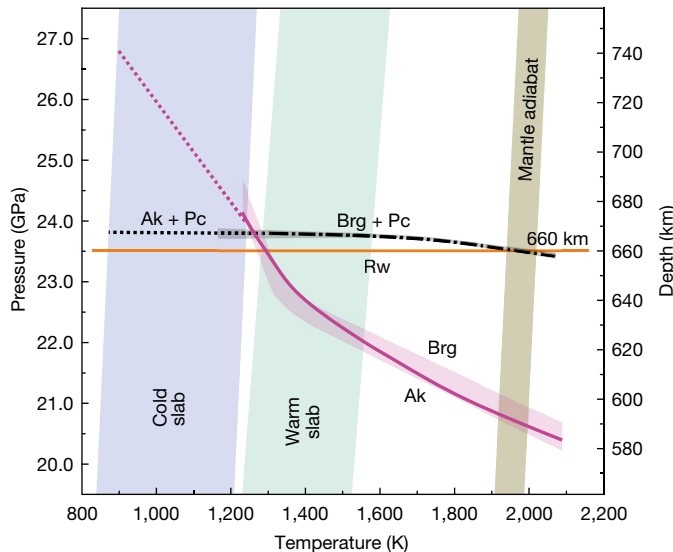

**Fig. 3 | Comparison of the RBP and AB phase transition boundaries determined in the present study.** Cold and warm subduction geotherms are from ref. [36,37] and the average mantle geotherm is after ref. [49]. RBP, dot-dashed black curve and grey shaded area; AB, solid violet curve and violet shaded area. The solid orange line is 660 km depth[50]. Ak, MgSiO₃ akimotoite; Brg, MgSiO₃ bridgmanite; Pc, MgO periclase; Rw, Mg₂SiO₄ ringwoodite. Dotted lines are the result of linear extrapolation of the RBP boundary below 1,500 K and the AB boundary below 1,290 K.

occupy up to 15% of Mg sites, leading to cation disorder. The effect of cation disorder on the heat capacity in ilmenite-structured minerals has not been investigated, but disorder causes an increase in entropy, leading to a curved phase boundary. Detailed studies regarding Ak cation disorder at high $P$–$T$ are required to understand the variation of the Clapeyron slope of the AB boundary.

## Topography of the D660

The present results predict the topography of the D660, especially its depression, beneath subduction zones. At $T < 1,260$ K, which is appropriate for cold subduction zones, the AB boundary is located at higher pressures than the extension of the RBP boundary (Fig. 3). Under this circumstance, Rw should first dissociate not to Brg plus fPc but to Ak plus fPc (the RAP transition), and Ak should then transform to Brg at higher pressures. In other words, the RBP transition should not be present in cold subduction zones, and the AB transition should cause the depressed D660, as proposed in previous theoretical[26,27] and seismological[19] studies. If the temperature is as low as 900 K, which some geodynamic studies[36,37] have predicted for cold subducted slabs, the D660 will be depressed to a depth of 740 km. Thus, the combination of the RAP and AB transitions will produce a double D660 at 660–740 km depths. The seismic wave velocity contrasts associated with these transitions are 40% and 60% from those associated with the RBP transition, respectively, as shown in Extended Data Fig. 3, which should be sufficiently large to detect each of them separately by seismological observations. At $T > 1,260$ K, which is appropriate for warm subduction zones, the RBP boundary should cause the D660 at approximately 660 km depth (Fig. 3).

The seismological observations of cold and warm slabs support the above predictions. One example is the very old (110 Myr[37]) Tonga slab, for which the calculated temperature at depths of 660–740 km is 870–1,270 K in the central part of the slab[37]. Seismological observations revealed a double D660 at depths of 738 km and 669 km for this slab[10], which can be interpreted as the AB and RAP transitions, respectively. A contrasting example is the Peru slab, which is younger (41 Myr; ref. [37]) and therefore warmer. The calculated temperature of the Peru slab

at approximately 660 km depth is 1,270–1,570 K in the central part of the slab[37], and seismological observations revealed the D660 at 656 km (ref. [15]), which can be interpreted as the RBP transition. Thus, the D660 depths calculated based on the AB, RAP and RBP boundaries and expected temperatures agree with the seismological observations beneath subduction zones. The D660 is globally observed near 660 km depth except for subduction zones, which should correspond to the RBP transition located at 23.4–23.8 GPa. We note that a depression of the D660 down to 690 km depth is observed in a hot plume[38], which neither the RBP or AB transition can explain. This observation can be interpreted as being caused by another phase transition, probably the garnet (Grt)–Brg phase transition[20]. Further study of this phase transition following our strategy is required to understand the origin of the D660 depression in the hot plume.

## Influence of Al and Fe

Although we did not investigate the effects of Al and Fe on the phase boundaries, we can predict their effects on the geophysical applications of our results. Because Ak and Brg form by the dissociation of Rw, they will contain no Al. When no Al is present, Fe strongly partitions into fPc, but neither Ak[39] nor Brg[40]. Our previous study also demonstrated that the pressures of Fe-free and Fe-bearing RBP transitions at 1,700 K are almost identical[1]. Although the Ak formed from Rw in cold slabs could react with Al-rich Grt to incorporate Al into Ak during further slab sinking (at 24–27 GPa and 1,100–1,250 K), elemental diffusivity should be too low at such temperatures to allow compositional equilibrium in Grt and Ak. Thus, the AB transition in cold slabs should essentially occur in the MgO–SiO₂ system, as is the case in our study.

## Geodynamic implications

Several geodynamics studies have shown that steep phase boundaries produce large buoyancy forces[41,42]. When the upward buoyancy produced by a negative phase boundary exceeds the downward force due to thermal expansion, mantle flows are hampered and slabs stagnate above the D660[43]. The boundary values of the Clapeyron slope, below which the slab will stagnate above the D660, were calculated as −6 MPa K⁻¹ (ref. [42]) and −3 MPa K⁻¹ (ref. [44]) in different geodynamics studies. Seismic tomography has revealed that warm young slabs (for example beneath Peru, the Marianas and Central America) penetrate the lower mantle[15,45], whereas cold old slabs (for example beneath the Izu–Bonin region, South Kurile and Japan) stagnate above D660[46]. The extremely steep negative Clapeyron slope of the AB phase transition at low temperatures (−8.1 MPa K⁻¹) will cause strong upward buoyancy and stagnation of cold subducted slabs, although the nearly neutral Clapeyron slope of the RBP phase transition at $T > 1,260$ K (−0.1 MPa K⁻¹) will allow the downward movement of warm slabs by thermal expansion.

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

## Methods

### Preparation of starting materials

$MgSiO_3$ bridgmanite and a mixture of $MgSiO_3$ bridgmanite and MgO periclase were used as starting materials for the determination of the AB and RBP phase transition boundaries, respectively. The starting material of bridgmanite was synthesized from enstatite powder at a pressure of 27 GPa and temperature of 1,300 K for 1 h. The mixture of bridgmanite + periclase was prepared using forsterite powder at a pressure of 27 GPa and temperature of 1,200 K for 5 min. The starting powders of enstatite and forsterite prepared by ref. [3] were used. High-pressure experiments to prepare the starting materials were conducted using a 15-MN Kawai-type multi-anvil press with the Osugi module at the Bayerisches Geoinstitut (BGI), University of Bayreuth (IRIS-15)[1,51]. Tungsten carbide (WC) anvils of grade TF05 (Fujilloy Co. Ltd) with a truncated edge length of 3.0 mm were used to compress the high-pressure cell. Pyrophyllite gaskets were used to seal the compressed volume and generate quasi-hydrostatic pressures inside the cell. A Cr-doped MgO 7-mm octahedron was used as a pressure medium. A Mo foil with a 25-μm thickness was used as a sample container and heating element. The furnace was surrounded by a $ZrO_2$ thermal insulator. Mo rods were placed at both ends of the heater to connect the heater/capsule with the WC anvils. The temperature was monitored on the surface of the heater with a $W_{97}Re_3$–$W_{75}Re_{25}$ thermocouple.

Reagent-grade MgO was used as a pressure marker for the in situ X-ray diffraction experiments and was sintered at 2 GPa and 770 K for 1 h using the 10-MN Kawai type multi-anvil press at BGI (Hymag). Tungsten carbide anvils with 15-mm truncated edge lengths were used to generate high pressure together with a Cr-doped MgO octahedron with a 25-mm edge length as the pressure medium. The inside of the pressure medium consisted of a stepped cylindrical graphite heater and $ZrO_2$ thermal insulator. A Mo foil (25-μm) was used as the sample capsule. The temperature was measured on the surface of the capsule using a $W_{97}Re_3$–$W_{75}Re_{25}$ thermocouple.

The recovered samples were analysed using a micro-focused X-ray diffractometer (Brucker AXS Discover 8) with a two-dimensional solid-state detector (VANTEC500) and micro-focus source (IμS) with Co-Kα radiation operated at 40 kV and 500 μA. We found a small amount of stishovite together with bridgmanite (Extended Data Fig. 4), which was also observed later in the diffraction spectra at high pressures and temperatures (Fig. 1b). A small amount of stishovite can be explained as excess $SiO_2$ cristobalite in the starting material (enstatite powder). The synthesized samples and sintered MgO were cut into half-disks of 0.8 mm in diameter and 0.5 mm in thickness.

### In situ X-ray diffraction experiments

The AB and RBP phase transition boundaries were determined at the DESY synchrotron radiation facility (Hamburg, Germany) and SPring-8 synchrotron radiation facility (Hyogo Prefecture, Japan), respectively. The experiments were performed at beamline P61b at DESY using the 3 × 5-MN six-axis multi-anvil press, and at beamline BL04B1 at SPring-8 using the 15-MN Kawai-type multi-anvil press, SPEED-*Mk*.II. The press loads of the six-axis press were the press loads of one axis multiplied by three, which corresponds to the press load of uniaxial presses such as SPEED-*Mk*.II. X-ray diffraction patterns were collected for 150–300 s for the pressure marker and 600–7,200 s for the sample using a Ge solid-state detector (SSD) with a 4,096-channel analyser. The SSD analyser was calibrated using the X-ray fluorescence lines of [55]Fe, [57]Co and [133]Ba before the measurements. The diffraction angle (2θ) was calibrated before each experiment with a precision of 0.0003° using MgO as a standard.

We used the same type of high-pressure cell described in ref. [1]. The high-pressure cell contained a $Cr_2O_3$–MgO pressure medium, cylindrical $LaCrO_3$ heater, $ZrO_2$ thermal insulator sleeve and Ta electrodes.

The sample and pressure marker with a half-disk shape were located in a 25-μm-thick Re cylindrical foil in the center of the experimental cell. Diamond/epoxy and boron/epoxy rods were put on both sides of the sample to minimize X-ray absorption. The sample temperature was measured on the surface of the Re capsule using a $W_{97}Re_3$–$W_{75}Re_{25}$ thermocouple. The thermocouple was isolated from the heater by $Al_2O_3$ insulator tubes. The pressure effect of the thermoelectromotive force of the thermocouple was corrected using the equations determined by ref. [48] after the experiments (Supplementary Tables 1 and 2).

The incident X-ray beam collimated to dimensions of 30–50 μm horizontally and 200–300 μm vertically was directed at the sample through the gaps between the second-stage anvils. All of the experiments were carried out with a press oscillation around the vertical press axis between 0° and 6° during the X-ray diffraction measurement to suppress intensity heterogeneities of the diffracted peaks due to possible grain growth at high temperature[52]. The pressure was obtained from the MgO unit cell volumes using the equations of state proposed by ref. [47] based on the third-order Birch–Murnaghan (3BM) and Vinet equations of state. To calculate the MgO unit-cell volumes, we usually used eight diffraction peaks (111, 200, 220, 311, 222, 400, 420, 422), which produce relatively high precision in pressure[1,3].

Although the experiments were carried out at different synchrotron facilities, the applied techniques were identical, with minor differences, which are described in detail in the next section (the AB boundary and the RBP boundary) and did not affect the overall result of the study.

### Determination of phase boundaries

As argued in refs. [1,3], the procedure to determine the phase boundaries in this study is based on the following ideas.

1. The starting materials are very reactive owing to the high-density defects produced during cold compression. The high-density defects can be inferred from the peak broadening during compression.
2. A phase that forms in situ is much more inert than the starting material due to low-defect density, which is inferred from the sharper diffraction peaks.
3. The phase that forms in situ is still reactive unless it is annealed at high temperatures for a long duration. This is likely because of the small grain size, which is inferred from the sharpening of diffraction peaks with increasing temperature.
4. It is difficult to form another new phase from a phase that forms in situ, probably because the surface energy hampers the formation of new grains. On the other hand, it is relatively easy to allow one phase to increase by consuming the other phase if the two phases are already present.
5. Reaction rates decrease when approaching the phase boundaries. It is therefore possible to maintain the coexistence of higher- and lower-pressure phases if the sample $P$–$T$ conditions are held close to the phase boundaries.
6. The reaction rate from one phase to the other decreases by annealing at high temperatures. This is likely because, although the transition from one phase to the other occurs through grain-boundary diffusion, the grain-boundary density should decrease due to grain growth by annealing.

For these reasons, we achieved coexistence of the higher- and lower-pressure phases at as low a temperature as possible at the beginning of the experiment. The sample temperature was then increased stepwise, and the phase boundaries were bracketed by small proceedings of the forward and reverse reactions in which the sample pressure was held near the phase boundary.

**The AB boundary.** The starting materials of bridgmanite were initially compressed to press loads of 10–12 MN at room temperature, at which the sample pressures were 30–32 GPa. They were then gradually heated to temperatures of 1,000–1,200 K, during which the sample

pressure spontaneously decreased to 24–26 GPa. At temperatures of 1,100–1,150 K, the starting materials of pure bridgmanite partially transformed into akimotoite. The temperature and press load were then maintained and diffraction patterns of the samples and pressure markers were successively collected. In many cases, the sample pressure spontaneously decreased. A stable phase was determined based on the relative increase or decrease of the diffraction peak intensities of the higher-pressure (Brg) to lower-pressure (Ak) phases. We typically used 6 Ak peaks (012, 104, 110, 113, 024, 116) and 15 Brg peaks (002, 110, 111, 020, 112, 200, 120, 210, 022, 202, 113, 122, 212, 220, 023) to estimate the diffraction peak intensities change. The change in the intensities of such a large number of diffraction peaks is associated with a quantitative fraction volume change of the coexisting phases. On the other hand, using only one or few peaks could be easily associated with grain growth or preferred orientation development, and therefore the reliability in such cases would be questionable. By crossing the phase transition boundary, the Brg/Ak ratio initially increased but then decreased. After the temperature and pressure stabilized, we considered the samples to be in the Brg stability field if the Brg/Ak ratio increased, but in the Ak stability field if the ratio decreased. If the spontaneous pressure decrease seemingly terminated, the press load was reduced slightly (by 0.05–0.15 MN) while maintaining the temperature to enhance the pressure drop until the Brg/Ak intensity ratio started to decrease. The lowest pressure at which the Brg/Ak ratio increased and highest pressure at which the ratio decreased were used to obtain the transition pressures. After the transition pressures were determined at this temperature, the sample temperature was increased by 50–100 K, and the above procedure was repeated to bracket the transition pressures at the higher temperature. If the Brg/Ak ratio decreased at the beginning of a new temperature stage, the sample temperature was reduced by 50–100 K, the press load was increased by 0.5–1.5 MN, and the sample was heated to the original temperature to enter the Brg stability field. After the samples were exposed at 2,085 K (2,000 K before correction), they were quenched and decompressed to ambient pressure.

**The RBP boundary.** The RBP boundary was determined in a similar way to the AB boundary. The procedural differences between these two boundaries are as follows. The mixtures of bridgmanite and periclase (starting material) were partially converted to ringwoodite at 1,200 K. The stable phase at high $P–T$ conditions was estimated mainly by the relative intensities of bridgmanite and ringwoodite diffraction peaks due to the relatively small proportion of periclase (Fig. 1a). We usually used 7 Rw peaks (220, 311, 222, 400, 422, 511, 333) and 18 Brg peaks (002, 110, 111, 020, 112, 120, 210, 121, 202, 113, 122, 212, 220, 023, 221, 130, 131, 311) to estimate the diffraction peak intensity changes. Because the total peak number is even larger than that used for the AB transition determination, the intensity changes of the diffraction peaks are also associated with a quantitative fraction volume change of the coexisting phases. The maximum experimental temperature was 2,040 K (1,950 K before correction).

## Data availability

The X-ray diffraction data that support the findings of this study are available on Zenodo (https://doi.org/10.5281/zenodo.5532573). Source data are provided with this paper.

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

**Acknowledgements** We thank H. Fischer, S. Übelhack, R. Njul, U. Trenz and S. Linhardt at the Bayerisches Geoinstitut and S. Sonntag at DESY for their technical assistance, K. Glazyrin for insightful discussions, and E. Posner for English correction of the manuscript. This work was funded by a research project approved by the Federal Ministry of Education and Research (BMBF) (05K16WC2), the European Research Council (ERC) under the European Union's Horizon 2020 research and innovation programme (proposal no. 787 527), the German Research Foundation (DFG) (KA3434/9-1) to T.K., and DFG (IS350/1-1) to T.I. The synchrotron X-ray diffraction experiments were performed at the PETRA III beamline P61B at DESY (Hamburg, Germany), a member of the Helmholtz Association HGF, and at the beamline BL04B1 at SPring-8 with the approval of the Japan Synchrotron Radiation Research Institute (JASRI) (proposal nos. 2019B1133, 2019A1353, 2018B1209, 2018B1218 and 2017B1078).

**Author contributions** A.C. and T.I. conducted experiments for the determinations of AB and RBP phase-transition boundaries, respectively. The current experimental strategy was developed by T.K. The experimental method was developed by T.I. T.I. instructed A.C. about the experimental method. D.B., Z.L., L.W., A.N., B.Y., H.T., Z.C., Y.H. and Y.T. operated the synchrotron radiation experiments at beamline BL04B1 at SPring-8. D.B., S.B., E.J.K. and R.F. operated synchrotron radiation experiments at beamline P61B at DESY. A.C. and T.I. analysed the data. A.C. wrote the first draft. All authors commented on the manuscript. The whole project was directed by T.K.

**Funding** Open access funding provided by Deutsches Elektronen-Synchrotron (DESY).

**Competing interests** The authors declare no competing interests.

**Additional information**
**Correspondence and requests for materials** should be addressed to Artem Chanyshev or Takayuki Ishii.

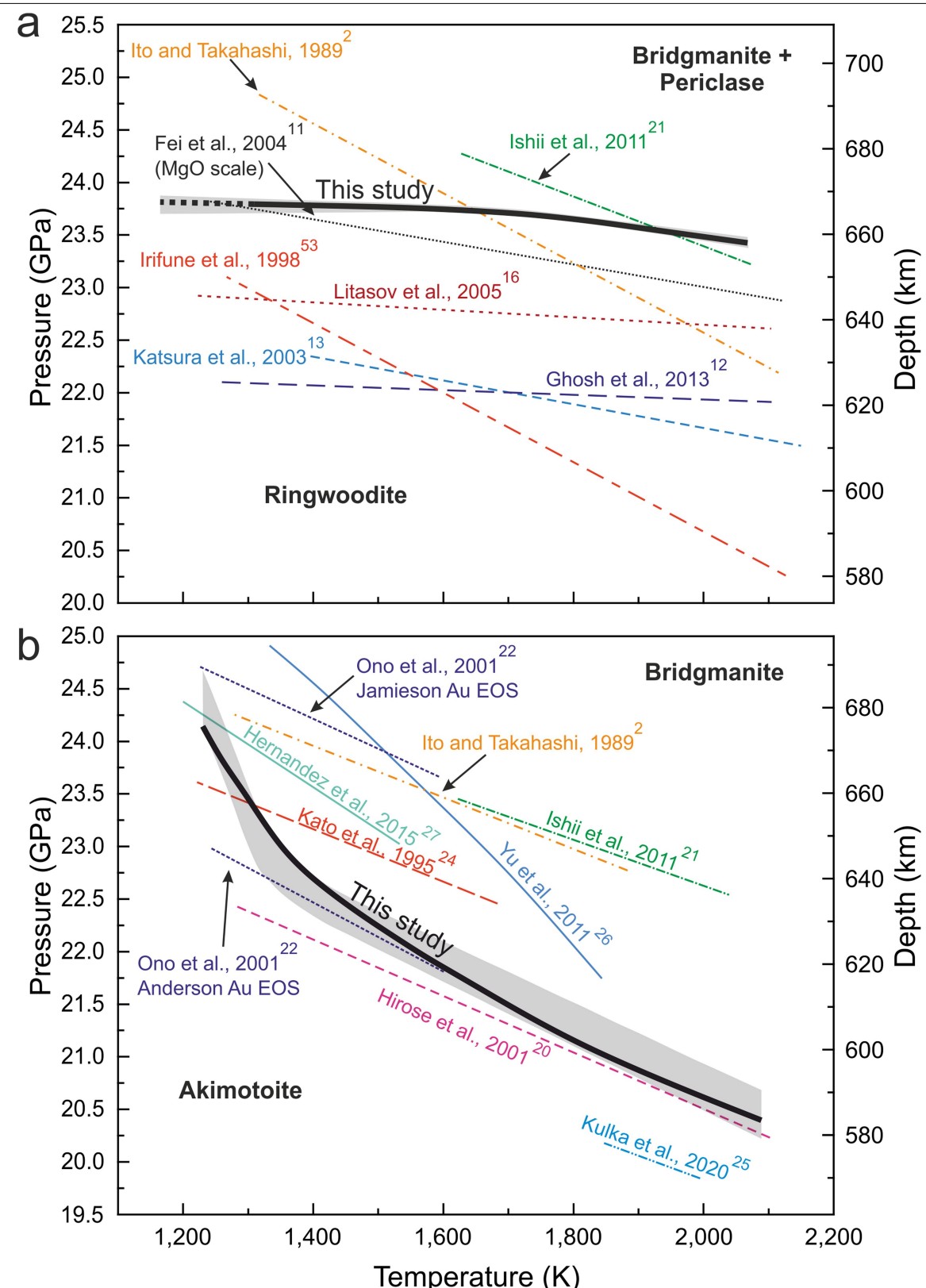

**Extended Data Fig. 1 | Comparison of the present results with previous data. a**, RBP transition boundary. **b**, AB transition boundary. The present data are shown with solid black lines and grey translucent areas. Phase transition boundaries determined in in situ X-ray multi-anvil experiments[11–13,16,20,22,53] are shown by dashed and dotted lines, those determined in multi-anvil laboratory experiments[2,21,25] are shown by dashed lines with dots, and those determined computationally[26,27] are shown by thin solid lines.

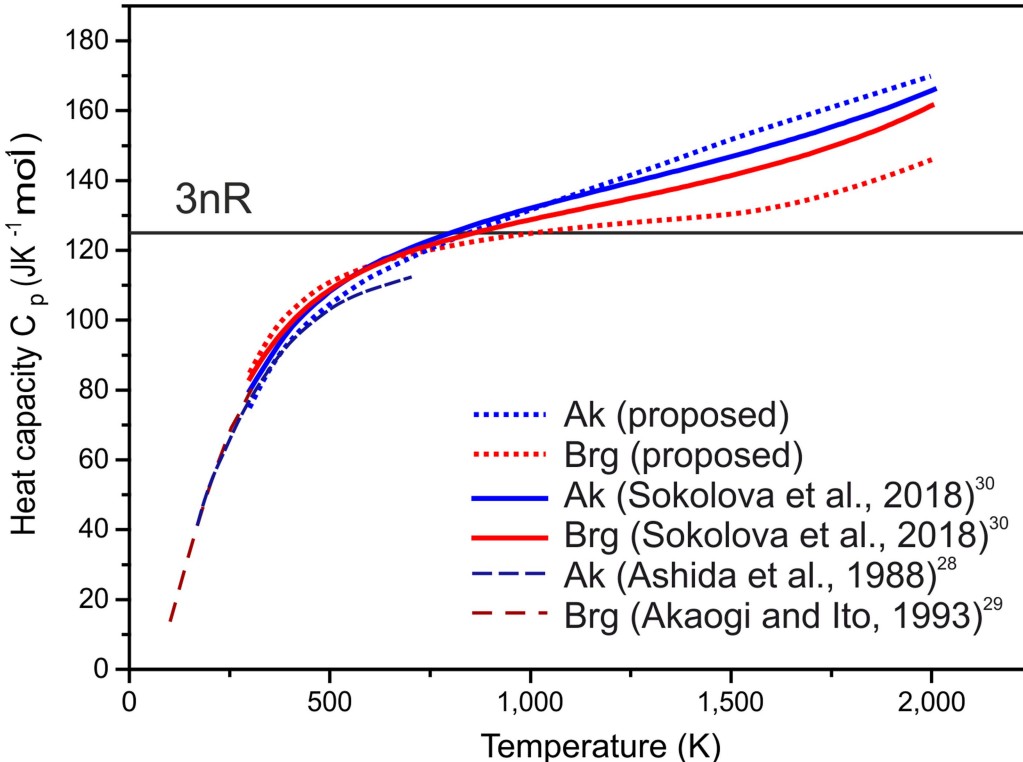

**Extended Data Fig. 2 | Heat capacities of MgSiO₃ Ak and Brg.** Comparison of the heat capacities of MgSiO₃ Ak and Brg explaining the present AB boundary (dotted blue and red lines) with those experimentally measured (thick solid blue and red lines)[28,29] and those calculated (dashed blue and dark red lines)[30]. 3nR denotes the limiting value of the Debye heat capacity.

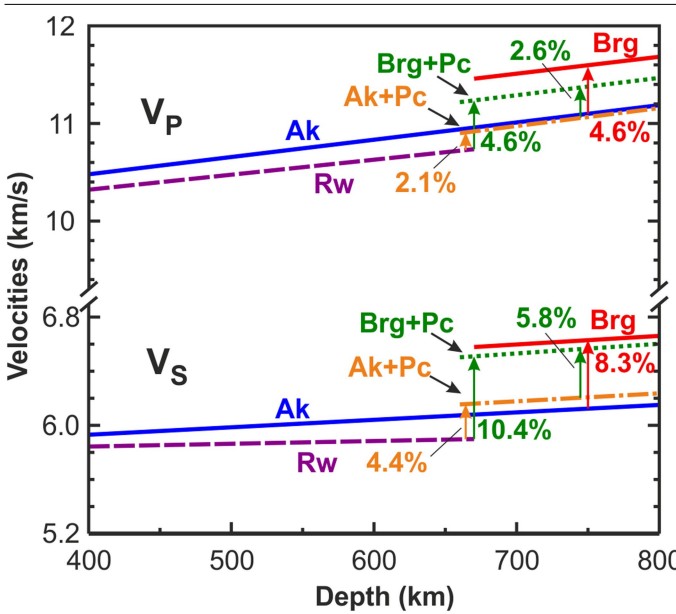

**Extended Data Fig. 3 | Seismic velocity contrasts caused by phase transitions.** $\Delta V_p$ and $\Delta V_s$ caused by the AB (red and blue solid lines and red arrows), RAP (purple dashed and orange dot-dashed lines and orange arrows), RBP (purple dashed and green dotted lines and green arrows), and Ak + Pc → Brg + Pc (orange dot-dashed and green dotted lines and green arrows) transitions along the cold geotherm. Data sources: Ak, ref. [54]; Rw, ref. [55]; Brg, ref. [56]; Pc, ref. [57]. Figure redrawn from ref. [54].

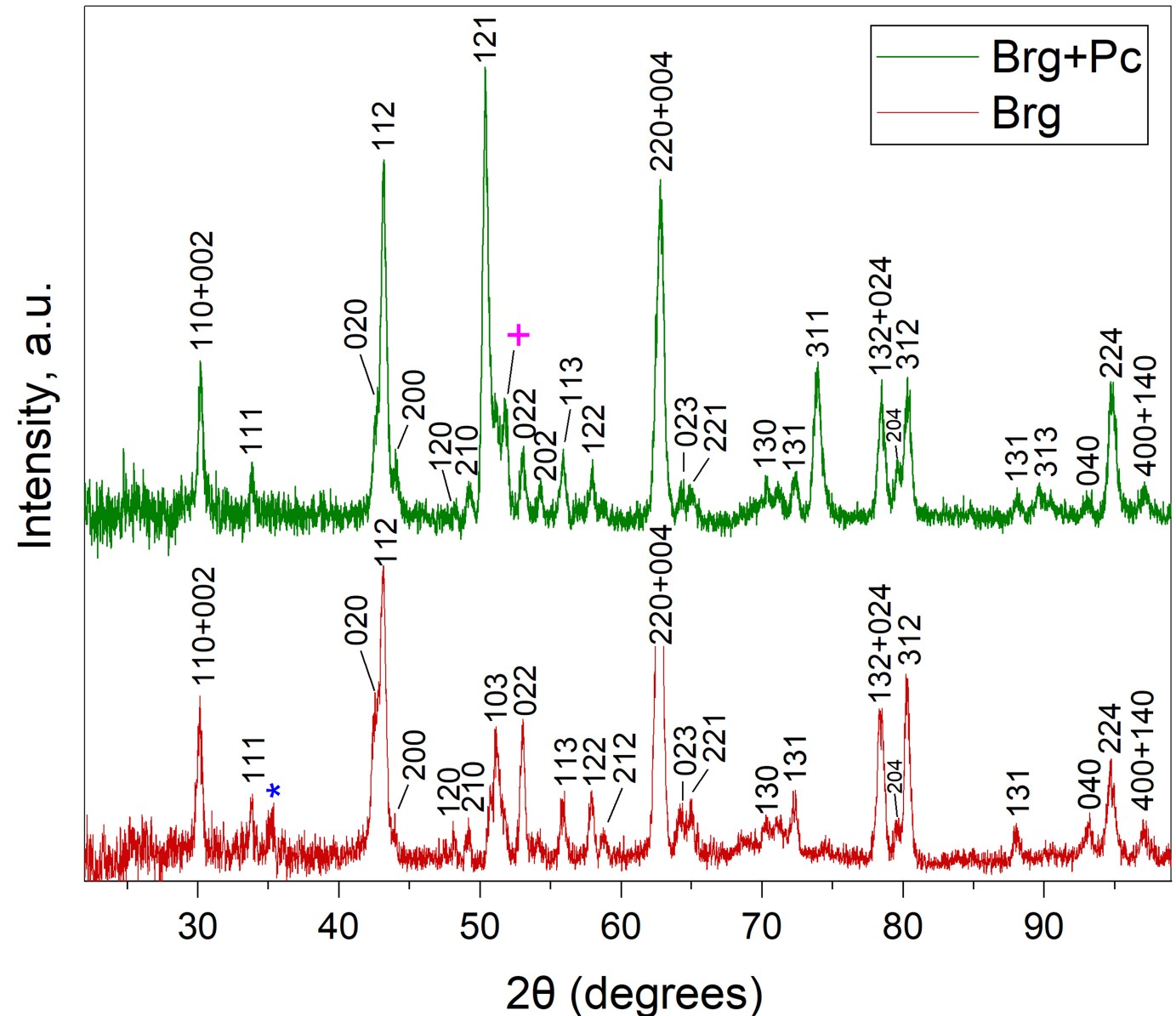

**Extended Data Fig. 4 | X-ray diffraction patterns of starting materials.** X-ray diffraction patterns of bridgmanite and bridgmanite plus periclase aggregates used as starting material to determine the AB and RBP phase transition boundaries, respectively. The samples were synthesized at 27 GPa and 1,000 K for 1 h in a multi-anvil press from enstatite powder to obtain bridgmanite, and from forsterite powder to obtain the mixture of bridgmanite and periclase. The diffraction peaks of stishovite and periclase are marked by a blue asterisk and pink plus, respectively.

**Extended Data Table 1 | Compressibility and thermal expansion coefficients of akimotoite and bridgmanite**

|  | Ak | Brg |
|---|---|---|
| *$K_{T0}$, GPa | $219.4^{(1)}$ | $261^{(3)}$ |
| †$K_{T0}'$ | $4.6^{(1)}$ | $4.0^{(3)}$ |
| ‡$V_0$, $cm^3 \cdot mol^{-1}$ | $26.35^{(1)}$ | $24.45^{(3)}$ |
| §$a_0$, $10^{-5}K^{-1}$ | $2.047^{(2)}$ | $1.982^{(3)}$ |
| §$a_1$, $10^{-9}K^{-2}$ | $4.00^{(2)}$ | $8.188^{(3)}$ |
| §$a_2$, $K^{-3}$ | $0.647^{(2)}$ | $0.474^{(3)}$ |

Data sources: (1) Zhou et al.[58]; (2) Wang et al.[59]; (3) Funamori et al.[60].

*Bulk modulus.

†Pressure derivative of the bulk modulus.

‡Molar volume ($cm^3 mol^{-1}$).

§Thermal expansion ($\alpha$) coefficients $\alpha = a_0 + a_1 {}^*T + a_2 {}^*T^{-2}$.