## [Peer Review File · Nature]

Manuscript Title: Depressed 660-km discontinuity caused by akimotoite-bridgmanite transition

Reviewer Comments & Author Rebuttals

Reviewer Reports on the Initial Version:

Referee #1:

This study employs newly refined experimental methods, resolving changing proportions of co-existing phases, to more precisely locate in pressure-temperature space the phase boundaries relevant to the seismologically observed 660 km discontinuity in the Earth's mantle. The most striking difference from previous studies of these phase equilibria (Extended Data Fig. 1) is the steep negative slope of the Akimotoite (Ak) → Bridgmanite (Brg) phase boundary at relatively low temperatures < 1500 K. It is this feature which underpins the novel implication of this study: attribution of the seismologically observed depression of the 660 km discontinuity by cold subducting slabs to the Ak-Brg phase transition. This finding will be of interest to a broad audience of earth scientists - including those in experimental petrology, mineral physics, seismology, and geodynamics. Accordingly, I consider that the manuscript is potentially acceptable for publication in Nature with an adequate response to the following two substantive suggestions.

Steeply negative Clapeyron slope for the Ak-Brg transition at low temperature

The markedly steeper negative Clapeyron slope of the Ak-Brg phase boundary at low temperature is thus critically important to the narrative presented in this manuscript. Accordingly, the authors offer some discussion at ll. 111-117 relating the 'third-law' entropy to specific heat. They suggest that an anomalous difference between the two phases in moderate-temperature specific heat $CP(T)$ is required to explain the temperature dependence of the Clapeyron slope. However, the authors should go further and offer the reader a possible reason for this marked anomaly.

The requirement that $\Delta S_{Ak \rightarrow Brg} = S_{Brg} - S_{Ak}$ be positive and relatively large at low temperature, but decrease with increasing temperature, could result from temperature dependent cation disorder. In particular, a gradual transition with increasing temperature from an Akimotoite structure in which Mg and Si are ordered to one with cation disorder, could enhance the increase in entropy with increasing temperature. Temperature-dependent cation disorder in Akimotoite (and at higher temperatures in Bridgmanite?) could also help explain the subtle variation in the slope of the phase boundary (Fig. 2) which is apparently flatter between 1500 and 1650 K?

The role of metastability within cold slabs

The discussion presented by the authors is based on the assumption that the seismologically observed 660 km discontinuity is associated with the location in P-T space of a phase boundary at which low- and high-pressure phases are in thermodynamic equilibrium. However, another possibility that has been widely discussed for cold slabs is the metastable persistence of the low-pressure phase assemblage to depths greater than those associated with thermodynamic equilibrium. The authors need to discuss this alternative explanation for the depression of the 660 km discontinuity.

Minor suggestions

I. 55: delete 'not' to avoid the effective double negative?

II. 138-141: please rewrite to make your meaning clearer.

Ian Jackson

21 January 2021

Referee #2:

The difficulty in reconciling seismic data for depth variations of the 660-km discontinuity with laboratory measurements of the P-T slopes of the relevant phase boundaries has long been a vexing problem in global geophysics. This paper tackles the problem with new measurements of two key boundaries (RBP and AB). In contrast to all previous studies, the authors find that the akimotoite-bridgmanite (AB) boundary steepens strongly at lower temperatures, providing a plausible explanation for the strong depression of the 660-km discontinuity seen in subduction zones. The experimental data reported here appear to be of high quality and carefully analyzed. The results would be of broad interest to Earth scientists. However, I have a number concerns about the manuscript as outlined below.

line 81-82 The authors argue that a key advantage of their approach is to constrain the phase boundary by the relative change in the diffraction peaks of the coexisting phases, since complete transformation from one high-P phase to another is not possible as described beginning from line 445. There are 2 points associated with this. First, this is not a new idea and has been used in previous studies – some citations are needed here. Second, this criterion can be ambiguous in the case of grain growth and development of preferred orientation. These are difficult to identify in energy dispersive experiments and it is not clear if press rotation is sufficient to counteract such effects. Further discussion of this issue is needed.

l119-121 The akimotoite - bridgmanite boundary is usually discussed in relation to the MgSiO_3 system. However, for the mantle (pyrolite composition) and subducting slabs (dominantly harzburgite and depleted pyrolite), it is the Mg_2SiO_4 system that dominates the mineralogy and is relevant for interpreting seismic discontinuities. The akimotoite to bridgmanite transition is not often considered in discussions of the Mg_2SiO_4 system, as the ringwoodite - akimotoite + ferropericlase reaction is restricted to quite low temperatures. To my knowledge, recent recognition of potential importance of this transition in an Mg_2SiO_4 -rich system comes mainly from theoretical studies (Yu et al 2011; Hernandez et al 2015) but these studies have not been cited here. But I may not be aware of all the literature here (Kuskov et al 89 also relevant?). Some clarification on this point is needed and those papers should be recognized.

A key result of this study is the change in Clapeyron slope of the AB phase transition at low T which would push the transition to deep depths in cold slabs. Most previous studies of phase transitions in the MgO-SiO_2 system produce sub-linear slopes (e.g., extended data fig. 1, for example). The authors suggest the non-linearity shown in Fig. 2b may be due to variations in the heat capacities of the phases and their effect on the corresponding entropy difference, but the current experimental data does not extend to sufficiently high temperature to address this (line 113). This isn't very satisfying. There are theoretical calculations that could be used to test this – e.g Hernandez et al 2015. While experimental data may be preferable, the theoretical results should not be ignored. The lack of a thermodynamic model to explain the slope change is a significant limitation of this work.

To explain a double seismic discontinuity in a subduction zone, sufficiently large velocity jumps are required across both transitions. While the Ak-Br transition would likely be expected to result in a significant increase in seismic velocities, it is less clear if the $\text{Rw} - \text{Ak} + \text{Pc}$ transition would result in an appreciable velocity increase as all three phases may have similar velocities around 25 GPa. The authors should use available elasticity data to provide an estimate of the velocity jumps across both of these transitions as a test of their suggestion that these two transitions are viable candidates to explain double discontinuities near 660.

An important point is that the authors should give credit to the work of Cottar and Deuss 2015 who previously proposed (based in part on theoretical studies cited above) that decomposition of

ringwoodite to akimotoite and periclase and akimotoite's subsequent transformation to bridgmanite could explain the depression of 660 in subduction zones. While the present study places the results on firmer footing through direct experiments, it is essential to give credit to those who first proposed this idea.

As illustrated in Fig. 3, the depth of the Ak+Pc transition would be expected to be substantially deeper in cold slabs than hot slabs. Is there evidence from seismic studies for this? This presents the possibility to provide a confirmation or refutation of the present proposal.

In the author contribution section, it is noted that the two phase transitions were studied by different scientists (presumably at different synchrotron facilities?). This is less than ideal. It is possible that different workers used different experimental or analysis techniques that might create some systematic difference between the results for the two materials. More clarification on this is needed.

The equation of state parameters chosen for akimotoite are outdated and likely incorrect. The study of Wang et al (2004) yielded anomalously large value for dK/dP relative to other high-pressure silicates. More recent experiments on akimotoite (although isentropic, not isothermal) indicate the values should be lower (Zhou et al 2014) which is consistent with the most advanced theoretical calculations (Hao et al 19; Karki and Wentzcovitch 02). It is not clear that this will have much effect on the results, but the better value should be used.

Minor points

Heading of Extended Table 3: change to "...after temperature correction due to thermocouple emf"

Line 227 – should be "blue asterisk"

Author Rebuttals to Initial Comments:

Reviewers Comments:

Referee #1 (Remarks to the Author):

Major comments:

1. The markedly steeper negative Clapeyron slope of the Ak-Brg phase boundary at low temperature is thus critically important to the narrative presented in this manuscript. Accordingly, the authors offer some discussion at ll. 111-117 relating the 'third-law' entropy

to specific heat. They suggest that an anomalous difference between the two phases in moderate-temperature specific heat $CP(T)$ is required to explain the temperature dependence of the Clapeyron slope. However, the authors should go further and offer the reader a possible reason for this marked anomaly.

The requirement that $\Delta S_{Ak \rightarrow Brg} = S_{Brg} - S_{Ak}$ be positive and relatively large at low temperature, but decrease with increasing temperature, could result from temperature dependent cation disorder. In particular, a gradual transition with increasing temperature from an Akimotoite structure in which Mg and Si are ordered to one with cation disorder, could enhance the increase in entropy with increasing temperature. Temperature-dependent cation disorder in Akimotoite (and at higher temperatures in Bridgmanite?) could also help explain the subtle variation in the slope of the phase boundary (Fig. 2) which is apparently flatter between 1500 and 1650 K?

We thank the reviewer for raising this critical point and for providing such a logical explanation. First, we have reconsidered our interpretation of the experimental data (Figs. 2 and 3 in the main text). We now indicate the P - T areas for both Rw-Brg+Pc and Ak-Brg transitions to account for the experimental data and pressure measurement errors, and propose a more smooth transition-boundary curve for the Ak-Brg boundary within the area that passes through the majority of the data points. The new interpretation suggests a gentler (-8.1 MPa/K) slope of the Ak-Brg boundary at low temperatures up to 1350 K, but a steeper slope (-3.2 MPa/K) above 1600 K than the previous interpretation (-21.2 and -2.1 MPa/K, respectively). This improvement provides a more realistic explanation for the entropy change. We consider the possible Ak and Brg heat capacities (ΔC_P) with temperature, which are similar to several minerals and generally agree with those calculated in a computational study (Sokolova et al., 2018) (Extended Data Fig. 2). In this evaluation, the Ak heat capacity should be lower than that of the Brg at $T < 860$ K but should become larger at higher temperatures. These temperature dependences could result from temperature-dependent cation disorder in Ak. On the other hand, the Brg heat capacity should not change significantly because the Rw-Brg+Pc boundary does not show a strong curvature at the same temperatures. The upward curved RBP boundary may suggest slight Mg–Si disorder at high temperatures. The heat capacity dependence of cation disorder in ilmenite-structure minerals has not been investigated, however, disorder causes an increase in entropy and accordingly heat capacity. Possible cation disorder in Ak was previously suggested in a few theoretical studies (Stixrude et al., 2003; Kiefer and Stixrude, 2002; Wentzcovitch et al., 2004), but was not experimentally examined by measuring Mg–Si disorder using X-ray or neutron diffraction. Nevertheless, a comparison of the measured distances of Mg–O (2.077 Å) and Si–O (1.799 Å) in Ak (Horiuchi et al., 1982) and Shannon's ionic radii of Mg (0.72 Å), Si (0.4 Å), and O (1.4 Å) (Ahrens 1952) implies that Si could occupy up to 15% of Mg-sites. Detailed studies regarding Ak cation disorder at high P - T are required to understand the variation of the Clapeyron slope of the AB boundary.

To address this concern, we have added an explanation of the possible theoretical justification for the Ak heat capacity change in the revised manuscript (L128–140). It is important to note that our main conclusion has not changed.

Below is a detailed argument regarding the above explanation.

The slope of a phase boundary is equal to the ratio of the entropy change (ΔS_{tr}) to the volume change (ΔV_{tr}) associated with the phase transition according to the Clausius-Clapeyron relation:

$$dP/dT = \Delta S_{tr}/\Delta V_{tr}.$$

The volume changes (ΔV_{tr}) of the AB transition vary only slightly over the entire investigated temperature range because both phases have similar thermal expansion coefficients and bulk moduli, and ΔV_{tr} can thus be estimated as $-1.7 \text{ cm}^3\text{mol}^{-1}$ (Yu et al., 2011) at low temperature and $-1.8 \text{ cm}^3\text{mol}^{-1}$ at high temperature. The entropy is a function of isobaric heat capacity (C_p) and temperature

$$S_T^0 = S_{298}^0 + \int_{T_0}^T \frac{C_p}{T} dT.$$

Therefore,

$$dP/dT \approx -0.5 * (\Delta S_{298}^0 + \Delta \int_{T_0}^T \frac{C_p}{T} dT).$$

ΔS_{298}^0 (Brg-Ak) was not determined unequivocally and varies from 4.13 (Yu et al., 2011) to 10 (Akaogi et al., 2002) $\text{JK}^{-1}\text{mol}^{-1}$. We used for our calculations $\Delta S_{298}^0 = 6.2$ (Yu et al., 2011) and **10** (Akaogi et al., 2002) $\text{JK}^{-1}\text{mol}^{-1}$.

$C_p = K_0 + K_1 T^{-0.5} + K_2 T^{-2} + K_3 T^{-3} + K_4 T + K_5 T^2 + K_6 T^3$, where $K_0, K_1, K_2, K_3, K_4, K_5$, and K_6 are constants.

Therefore,

$$\int_{T_0}^T \frac{C_p}{T} dT = K_0 \ln(T) - 2K_1 T^{-0.5} - 0.5K_2 T^{-2} - \frac{1}{3}K_3 T^{-3} + K_4 T + 0.5K_5 T^2 + \frac{1}{3}K_6 T^3.$$

Additionally, constants $K_0 - K_6$ could be different over different temperature ranges because:

$$S_{T_3}^0 = S_{298}^0 + \int_{T_0}^{T_1} \frac{C_p}{T} dT + \int_{T_1}^{T_2} \frac{C_p}{T} dT + \int_{T_2}^{T_3} \frac{C_p}{T} dT$$

Using these equations, we propose the following constants for Ak and Brg in the different temperature ranges to explain the gradually varied Ak-Brg boundary:

1. $\Delta S_{298}^0 = 6.2$;

Ak: 300–1300 K: $K_0 = 96.9$; $K_4 = 39.9 \times 10^{-3}$; $K_2 = -3.2 \times 10^{-6}$
 1300–2000 K: $K_0 = 117.0$; $K_4 = 25.0 \times 10^{-3}$; $K_2 = -3.0 \times 10^{-6}$

Brg: 300–1600 K: $K_0 = 124.2$; $K_4 = 7.5 \times 10^{-3}$; $K_2 = -3.6 \times 10^{-6}$
 1600–2000 K: $K_0 = -168.56$; $K_1 = 5655$; $K_2 = -1.86 \times 10^7$; $K_3 = 2.02 \times 10^8$;
 $K_4 = 0.1835$; $K_5 = -7.33 \times 10^{-5}$; $K_6 = 1.56 \times 10^{-8}$

$\Delta V = -1.7(-1.8) \text{ cm}^3 \text{ mol}^{-1}$

$\Delta S_{298}^0 = 6.2 \text{ JK}^{-1} \text{ mol}^{-1}$

Ak: 300-1300 K: $C_p = 96.0 + 39.9 \times 10^{-3} T - 3.2 \times 10^{-6} T^2$

1300-2000 K: $C_p = 117.0 + 25.0 \times 10^{-3} T - 3.0 \times 10^{-6} T^2$

Brg: 300-1600 K: $C_p = 124.2 + 7.5 \times 10^{-3} T - 3.6 \times 10^{-6} T^2$

1600-2000 K: $C_p = -168.56 + 5655 T^{-0.5} - 1.86 \times 10^7 T^{-2} + 2.02 \times 10^8 T^{-3} + 0.1835 T - 7.33 \times 10^{-5} T^2 + 1.56 \times 10^{-8} T^3$

2. $\Delta S_{298}^0 = 10$;

Ak: 300–1400 K: $K_0 = 100.0$; $K_4 = 34.9 \times 10^{-3}$; $K_2 = -3.2 \times 10^{-6}$
 1400–2000 K: $K_0 = 101.0$; $K_4 = 34.9 \times 10^{-3}$; $K_2 = -3.2 \times 10^{-6}$
 Brg: 300–1400 K: $K_0 = 120.3$; $K_4 = 8.0 \times 10^{-3}$; $K_2 = -3.4 \times 10^{-6}$
 1400–1600 K: $K_0 = 118.0$; $K_4 = 9.0 \times 10^{-3}$; $K_2 = -3.0 \times 10^{-6}$
 1600–2000 K: $K_0 = -164.6$; $K_1 = 5655$; $K_2 = -1.86 \times 10^7$; $K_3 = 2.02 \times 10^8$;
 $K_4 = 0.1785$; $K_5 = -7.32 \times 10^{-5}$; $K_6 = 1.56 \times 10^{-8}$

$\Delta V = -1.7(-1.8) \text{ cm}^3 \text{ mol}^{-1}$
 $\Delta S_{298}^0 = 10 \text{ JK}^{-1} \text{ mol}^{-1}$
 Ak: 300-1400 K: $C_p = 100.0 + 34.9 \times 10^{-3} T - 3.2 \times 10^{-6} T^2$
 1400-2000 K: $C_p = 101.0 + 34.9 \times 10^{-3} T - 3.2 \times 10^{-6} T^2$
 Brg: 300-1400 K: $C_p = 120.3 + 8.0 \times 10^{-3} T - 3.4 \times 10^{-6} T^2$
 1400-1600 K: $C_p = 118.0 + 9.0 \times 10^{-3} T - 3.0 \times 10^{-6} T^2$
 1600-2000 K: $C_p = -164.6 + 5655 T^{-0.5} - 1.86 \times 10^7 T^{-2} + 2.02 \times 10^9 T^{-3} +$
 $0.1785 T - 7.32 \times 10^{-5} T^2 + 1.56 \times 10^{-8} T^3$

A gradual change of the Ak and Brg heat capacities with increasing temperature can therefore explain the gradually varied Ak-Brg transition boundary.

2. The role of metastability within cold slabs

The discussion presented by the authors is based on the assumption that the seismologically observed 660 km discontinuity is associated with the location in P-T space of a phase boundary at which low- and high-pressure phases are in thermodynamic equilibrium. However, another possibility that has been widely discussed for cold slabs is the metastable persistence of the low-pressure phase assemblage to depths greater than those associated with thermodynamic equilibrium. The authors need to discuss this alternative explanation for the depression of the 660 km discontinuity.

As the reviewer suggests, the metastable persistence of the low-pressure phase assemblage to greater depths could be a possible explanation of the discontinuity depressions. Tetzlaff and Schmelting (2000) proposed that the sluggish kinetics of the RBP phase transition at low temperatures (800–1000 K) could be the reason for the D660 depressions beneath cold subduction zones. Please note that a seismological study (Tibi and Wiens, 2005) showed that the D660 is extraordinarily sharp and less than 2-km thick in cold subduction zones. Although such sharpness can be interpreted by an overpressure-induced RBP transition only at extremely low temperatures (<1000 K, Kubo et al., 2002), this estimate is not convincing because the reaction rate of the RBP transition at such temperatures was determined with significant variations by extrapolation of the higher-*T* data (Kubo et al., 2002). Nevertheless, this explanation is also possible as we further discuss in the text (L58–63).

Minor suggestions

I. 55: delete ‘not’ to avoid the effective double negative?

Corrected. We have rewritten these sentences as “However, this suggestion could be invalid due to improper experimental procedure; the formation of ringwoodite from bridgmanite plus ferropericlase was not demonstrated under wet conditions in this study” (L56-58)

II. 138-141: please rewrite to make your meaning clearer.

Corrected. We have rewritten these sentences as: “Although the Ak formed from Rw in cold slabs could react with Al-rich Grt to incorporate Al into Ak during further slab sinking (at 24–27 GPa and 1100–1250 K), elemental diffusivity should be too low at such *T* to allow compositional equilibrium in Grt and Ak. Thus, the AB transition in cold slabs should essentially occur in the MgO-SiO₂ system, as was the case in our study.” (L176–180)

Referee #2 (Remarks to the Author):

line 81-82 The authors argue that a key advantage of their approach is to constrain the phase boundary by the relative change in the diffraction peaks of the coexisting phases, since complete transformation from one high-P phase to another is not possible as described beginning from line 445. There are 2 points associated with this. First, this is not a new idea and has been used in previous studies – some citations are needed here. Second, this criterion can be ambiguous in the case of grain growth and development of preferred orientation. These are difficult to identify in energy dispersive experiments and it is not clear if press rotation is sufficient to counteract such effects. Further discussion of this issue is needed.

Thank you for making these points.

Regarding the first point, we agree that the determination of the phase boundary by the relative change in the diffraction peaks of the coexisting phases is not a new idea. In the main text, we indicate that this idea was already applied by Litasov et al. (2005) (L91–94).

Regarding the second point, we theoretically agree with the reviewer that preferred orientation may have affected the determination of the phase relations because the openings of multianvil presses for X-ray accommodation and angle ranges and directions of the press oscillation are limited. We sometimes found that the magnitudes of the diffraction intensity changes varied among different peaks when the sample conditions were close to the possible phase boundaries. If the change in diffraction patterns caused such confusion, however, we excluded this data point from the data set. We adopted a data point only if some peak intensities of one phase increased, none of the same phase decreased, some of the other phase decreased, and none of the other phase increased (however, sometimes we observed a slight increase of the one peak intensity of the other phase – see Brg 220 in Fig. 1a and Brg 022 in Fig. 1b). We also consider that the majority of grains should not have disappeared or newly formed near the phase boundary and that already existing grains should have grown or been reduced by grain-boundary diffusion, judging from the small fraction changes. Hence, the general patterns of preferred orientation should not have changed. Furthermore, we compared more than two diffraction patterns during a spontaneous pressure decrease at constant temperature and press load, which led to no change of deviatoric stresses that would change the preferred orientation. The effects of preferred orientation on the phase boundary determinations were therefore likely minimal.

Line 119-121 The akimotoite - bridgmanite boundary is usually discussed in relation to the MgSiO₃ system. However, for the mantle (pyrolite composition) and subducting slabs (dominantly harzburgite and depleted pyrolite), it is the Mg₂SiO₄ system that dominates the mineralogy and is relevant for interpreting seismic discontinuities. The akimotoite to bridgmanite transition is not often considered in discussions of the Mg₂SiO₄ system, as the ringwoodite - akimotoite + ferropericlasite reaction is restricted to quite low temperatures. To my knowledge, recent recognition of potential importance of this transition in an Mg₂SiO₄-rich system comes mainly from theoretical studies (Yu et al 2011; Hernandez et al 2015) but these studies have not been cited here. But I may not be aware of all the literature here (Kuskov

et al 89 also relevant?). Some clarification on this point is needed and those papers should be recognized.

We have indicated that this idea has already been previously considered (L147).

A key result of this study is the change in Clapeyron slope of the AB phase transition at low T which would push the transition to deep depths in cold slabs. Most previous studies of phase transitions in the MgO-SiO₂ system produce sub-linear slopes (e.g., extended data fig. 1, for example). The authors suggest the non-linearity shown in Fig. 2b may be due to variations in the heat capacities of the phases and their effect on the corresponding entropy difference, but the current experimental data does not extend to sufficiently high temperature to address this (line 113). This isn't very satisfying. There are theoretical calculations that could be used to test this – e.g Hernandez et al 2015. While experimental data may be preferable, the theoretical results should not be ignored. The lack of a thermodynamic model to explain the slope change is a significant limitation of this work.

Thank you for raising this important point. Indeed, some theoretical studies reported calculated heat capacities at higher temperatures, e.g up to 1600 K (Hernandez et al., 2015) or up to 2000 K (Sokolova et al., 2018). Using these calculated values, the AB boundary should have a gradually steep curve with Clapeyron slopes = -3.5 ± 0.8 MPa/K, which contradicts our results. To obtain a good thermodynamic model for our results, we reconsidered our interpretation of the experimental data (Figs. 2 and 3 in the main text). We now indicate the P - T areas for both Rw-Brg+Pc and Ak-Brg transitions to account for the experimental data and pressure measurement errors, and propose a more smooth transition-boundary curve for the Ak-Brg boundary within the area that passes through the majority of the data points. The new interpretation suggests a gentler (-8.1 MPa/K) slope of the Ak-Brg boundary at low temperatures up to 1350 K, but a steeper slope (-3.2 MPa/K) above 1600 K than the previous interpretation (-21.2 and -2.1 MPa/K, respectively). This improvement provides a more realistic explanation for the entropy change. We consider the possible Ak and Brg heat capacities (ΔC_P) with temperature, which are similar to several minerals and generally agree with those calculated in a computational study (Sokolova et al., 2018) (Extended Data Fig. 2). In this evaluation, the Ak heat capacity should be lower than that of the Brg at $T < 860$ K but should become larger at higher temperatures. These temperature dependences could result from temperature-dependent cation disorder in Ak. On the other hand, the Brg heat capacity should not change significantly because the Rw-Brg+Pc boundary does not show a strong curvature at the same temperatures. The upward curved RBP boundary may suggest slight Mg-Si disorder at high temperatures. The heat capacity dependence of cation disorder in ilmenite-structure minerals has not been investigated, however, disorder causes an increase in entropy and accordingly heat capacity. Possible cation disorder in Ak was previously suggested in a few theoretical studies (Stixrude et al., 2003; Kiefer and Stixrude, 2002; Wentzcovitch et al., 2004), but was not experimentally examined by measuring Mg-Si disorder using X-ray or

neutron diffraction. Nevertheless, a comparison of the measured distances of Mg-O (2.077 Å) and Si-O (1.799 Å) in Ak (Horiuchi et al., 1982) and Shannon's ionic radii of Mg (0.72 Å), Si (0.4 Å), and O (1.4 Å) (Ahrens 1952) implies that Si could occupy up to 15% of Mg-sites. Detailed studies regarding Ak cation disorder at high P - T are required to understand the variation of the Clapeyron slope of the AB boundary.

To address this concern, we have added an explanation of the possible theoretical justification for the Ak heat capacity change in the revised manuscript (L128–140). It is important to note that our main conclusion has not changed.

Below is a detailed argument regarding the above explanation.

The slope of a phase boundary is equal to the ratio of the entropy change (ΔS_{tr}) to the volume change (ΔV_{tr}) associated with the phase transition according to the Clausius-Clapeyron relation:

$$dP/dT = \Delta S_{tr}/\Delta V_{tr}.$$

The volume changes (ΔV_{tr}) of the AB transition vary only slightly over the entire investigated temperature range because both phases have similar thermal expansion coefficients and bulk moduli, and ΔV_{tr} can thus be estimated as $-1.7 \text{ cm}^3\text{mol}^{-1}$ (Yu et al., 2011) at low temperature and $-1.8 \text{ cm}^3\text{mol}^{-1}$ at high temperature. The entropy is a function of isobaric heat capacity ($C_{p\downarrow}$) and temperature

$$S_T^0 = S_{298}^0 + \int_{T_0}^T \frac{C_p}{T} dT.$$

Therefore,

$$dP/dT \approx -0.5 * (\Delta S_{298}^0 + \Delta \int_{T_0}^T \frac{C_p}{T} dT).$$

ΔS_{298}^0 (Brg-Ak) was not determined unequivocally and varies from 4.13 (Yu et al., 2011) to 10 (Akaogi et al., 2002) $\text{JK}^{-1}\text{mol}^{-1}$. We used for our calculations $\Delta S_{298}^0 = 6.2$ (Yu et al., 2011) and **10** (Akaogi et al., 2002) $\text{JK}^{-1}\text{mol}^{-1}$.

$C_p = K_0 + K_1 T^{-0.5} + K_2 T^{-2} + K_3 T^{-3} + K_4 T + K_5 T^2 + K_6 T^3$, where $K_0, K_1, K_2, K_3, K_4, K_5$, and K_6 are constants.

Therefore,

$$\int_{T_0}^T \frac{C_p}{T} dT = K_0 \ln(T) - 2K_1 T^{-0.5} - 0.5K_2 T^{-2} - \frac{1}{3}K_3 T^{-3} + K_4 T + 0.5K_5 T^2 + \frac{1}{3}K_6 T^3.$$

Additionally, constants K_0 – K_6 could be different over different temperature ranges because:

$$S_{T_3}^0 = S_{298}^0 + \int_{T_0}^{T_1} \frac{C_p}{T} dT + \int_{T_1}^{T_2} \frac{C_p}{T} dT + \int_{T_2}^{T_3} \frac{C_p}{T} dT$$

nature portfolio

Using these equations, we propose the following constants for Ak and Brg in the different temperature ranges to explain the gradually varied Ak-Brg boundary:

1. $\Delta S_{298}^0 = 6.2$;

Ak: 300–1300 K: $K_0 = 96.9$; $K_4 = 39.9 \times 10^{-3}$; $K_2 = -3.2 \times 10^{-6}$
 1300–2000 K: $K_0 = 117.0$; $K_4 = 25.0 \times 10^{-3}$; $K_2 = -3.0 \times 10^{-6}$

Brg: 300–1600 K: $K_0 = 124.2$; $K_4 = 7.5 \times 10^{-3}$; $K_2 = -3.6 \times 10^{-6}$
 1600–2000 K: $K_0 = -168.56$; $K_1 = 5655$; $K_2 = -1.86 \times 10^7$; $K_3 = 2.02 \times 10^8$;
 $K_4 = 0.1835$; $K_5 = -7.33 \times 10^{-5}$; $K_6 = 1.56 \times 10^{-8}$

$\Delta V = -1.7(-1.8) \text{ cm}^3 \text{ mol}^{-1}$

$\Delta S_{298}^0 = 6.2 \text{ JK}^{-1} \text{ mol}^{-1}$

Ak: 300-1300 K: $C_p = 96.0 + 39.9 \times 10^{-3} \times T - 3.2 \times 10^{-6} \times T^2$

1300-2000 K: $C_p = 117.0 + 25.0 \times 10^{-3} \times T - 3.0 \times 10^{-6} \times T^2$

Brg: 300-1600 K: $C_p = 124.2 + 7.5 \times 10^{-3} \times T - 3.6 \times 10^{-6} \times T^2$

1600-2000 K: $C_p = -168.56 + 5655 \times T^{-0.5} - 1.86 \times 10^7 \times T^{-2} + 2.02 \times 10^8 \times T^{-3} + 0.1835 \times T - 7.33 \times 10^{-5} \times T^2 + 1.56 \times 10^{-8} \times T^3$

2. $\Delta S_{298}^0 = 10$;

Ak: 300–1400 K: $K_0 = 100.0$; $K_4 = 34.9 \times 10^{-3}$; $K_2 = -3.2 \times 10^{-6}$
 1400–2000 K: $K_0 = 101.0$; $K_4 = 34.9 \times 10^{-3}$; $K_2 = -3.2 \times 10^{-6}$
 Brg: 300–1400 K: $K_0 = 120.3$; $K_4 = 8.0 \times 10^{-3}$; $K_2 = -3.4 \times 10^{-6}$
 1400–1600 K: $K_0 = 118.0$; $K_4 = 9.0 \times 10^{-3}$; $K_2 = -3.0 \times 10^{-6}$
 1600–2000 K: $K_0 = -164.6$; $K_1 = 5655$; $K_2 = -1.86 \times 10^7$; $K_3 = 2.02 \times 10^8$;
 $K_4 = 0.1785$; $K_5 = -7.32 \times 10^{-5}$; $K_6 = 1.56 \times 10^{-8}$

$\Delta V = -1.7(-1.8) \text{ cm}^3 \text{ mol}^{-1}$
 $\Delta S_{298}^0 = 10 \text{ JK}^{-1} \text{ mol}^{-1}$
 Ak: 300-1400 K: $C_p = 100.0 + 34.9 \times 10^{-3} T - 3.2 \times 10^{-6} T^2$
 1400-2000 K: $C_p = 101.0 + 34.9 \times 10^{-3} T - 3.2 \times 10^{-6} T^2$
 Brg: 300-1400 K: $C_p = 120.3 + 8.0 \times 10^{-3} T - 3.4 \times 10^{-6} T^2$
 1400-1600 K: $C_p = 118.0 + 9.0 \times 10^{-3} T - 3.0 \times 10^{-6} T^2$
 1600-2000 K: $C_p = -164.6 + 5655 T^{0.5} - 1.86 \times 10^7 T^2 + 2.02 \times 10^8 T^3 +$
 $0.1785 T - 7.32 \times 10^{-5} T^2 + 1.56 \times 10^{-8} T^3$

A gradual change of the Ak and Brg heat capacities with increasing temperature can therefore explain the gradually varied Ak-Brg transition boundary.

To explain a double seismic discontinuity in a subduction zone, sufficiently large velocity jumps are required across both transitions. While the Ak-Br transition would likely be expected to result in a significant increase in seismic velocities, it is less clear if the Rw - Ak + Pc transition would result in an appreciable velocity increase as all three phases may have similar velocities around 25 GPa. The authors should use available elasticity data to provide an estimate of the velocity jumps across both of these transitions as a test of their suggestion that these two transitions are viable candidates to explain double discontinuities near 660.

We calculated the velocity jumps associated with the $Rw \rightarrow Ak + Pc$ and $Ak + Pc \rightarrow Brg + Pc$ transitions under cold slab conditions, as shown in Extended Data Figure 3. The magnitudes of the velocity jumps with these transitions are 2.1% and 2.6% in ΔV_p , respectively, and 4.5% and 5.8% in ΔV_s , respectively. Although the velocity jumps with $Rw \rightarrow Ak + Pc$ are thus smaller than those with $Ak + Pc \rightarrow Brg + Pc$, as the reviewer expected, the velocity jumps for these transitions are 40% and 60% of the RBP transition, respectively, and the $Rw \rightarrow Ak + Pc$ transition can therefore produce sufficiently large velocity jumps that can be detected by seismology (L150–153).

An important point is that the authors should give credit to the work of Cottaar and Deuss 2015 who previously proposed (based in part on theoretical studies cited above) that decomposition of ringwoodite to akimotoite and periclase and akimotoite's subsequent transformation to bridgmanite could explain the depression of 660 in subduction zones. While the present study places the results on firmer footing through direct experiments, it is essential to give credit to those who first proposed this idea.

Thank you for this correction. We indicated that this idea has been previously considered (L65–66).

As illustrated in Fig. 3, the depth of the Ak+Pc transition would be expected to be substantially deeper in cold slabs than hot slabs. Is there evidence from seismic studies for this? This presents the possibility to provide a confirmation or refutation of the present proposal.

As shown in Fig. 3, the Rw-Ak+Pc (RAP) transition occurs only with cold slab parameters at approximately 660-km depth. Here we clarify which phase transition can produce discontinuities under cold and warm slab conditions. According to our results, under cold slab conditions, the Rw-Ak+Pc transition should occur at ≈ 670 km and the Ak-Brg transition should occur at ≈ 670 – 750 km, whereas under warm slab conditions, the Rw-Brg+Pc transition should occur at ≈ 670 km and the Ak-Brg transition should occur at ≈ 620 – 640 km. However, the Ak amount in the transition zone under warm slab conditions should be insufficient (Ishii et al., 2011; 2019; Zhang et al., 2013) for the AB transition to be detected by seismology, and therefore only the Rw-Brg+Pc transition should occur under warm slab conditions.

The seismological observations of cold and warm slabs support our results. For example, the Tonga slab is extremely cold (870–1270 K in the central part at depths of 660–740 km), and

seismological observations revealed a double D660 at depths of 738 and 669 km for this slab (Zang et al., 2006), which can be interpreted by the AB and RAP transitions, respectively. A contrasting example is the Peru slab, which is warmer. The calculated T of the Peru slab at approximately 660-km depth is 1270–1570 K in the central part, and seismological observations revealed the D660 at 656 km (Fukao et al., 2013), which can be interpreted by the RBP transition. The D660 depths calculated based on the AB, RAP, and RBP boundaries and expected temperatures therefore agree with seismological observations beneath subduction zones (L156–165).

In the author contribution section, it is noted that the two phase transitions were studied by different scientists (presumably at different synchrotron facilities?). This is less than ideal. It is possible that different workers used different experimental or analysis techniques that might create some systematic difference between the results for the two materials. More clarification on this is needed.

Indeed, these two phase transitions were studied at different synchrotron facilities, as described in the Methods (L384–386). However, the applied techniques were identical with minor differences because the experimental stations at both facilities were built under the leadership of Tomoo Katsura with the same philosophy. The minor differences are described in detail in sections *The AB boundary* (L443–471) and *The RBP boundary* (L472–483). The experimental method was developed by Takayuki Ishii. Artem Chanyshv learned this method by direct instruction by Takayuki Ishii. Takayuki Ishii accompanied some runs conducted by Artem Chanyshv. Furthermore, the whole project was directed by Tomoo Katsura. Both of the experimental stations for multianvil experiments are constructed based on the same philosophy and under the leadership of Tomoo Katsura.

The equation of state parameters chosen for akimotoite are outdated and likely incorrect. The study of Wang et al (2004) yielded anomalously large value for dK/dP relative to other high-pressure silicates. More recent experiments on akimotoite (although isentropic, not isothermal) indicate the values should be lower (Zhou et al 2014) which is consistent with the most advanced theoretical calculations (Hao et al 19; Karki and Wentzcovitch 02). It is not clear that this will have much effect on the results, but the better value should be used.

Thank you for this correction. The values of the akimotoite bulk modulus and pressure derivative of the bulk modulus were replaced (L589–590). The new values are even closer to the values for bridgmanite, which confirms our conclusion (i.e., volume changes of the AB transition vary only slightly over the entire investigated temperature range).

Minor points

Heading of Extended Table 3: change to “...after temperature correction due to thermocouple emf”

Corrected (L606).

Line 227 – should be “blue asterisk”

Corrected (L584).

Reviewer Reports on the First Revision:

Referee #1:

The manuscript has been significantly improved in response to the reviewers' comments and is now acceptable for publication in Nature, in my opinion.

Referee #2:

The authors have carefully addressed all the points raised by both reviewers. As a result, the manuscript is now substantially improved. In response to a good suggestion from the other reviewer, the authors now argue that cation disorder changes with temperature in akimotoite may provide an explanation for the large slope change and the differences with theoretical calculations. This is possible but at present unproven, and will no doubt stimulate further work to confirm or disprove this idea. Overall, an impressive paper and I support publication. The work will be of strong interest to mineral physicists, seismologists, and geodynamicists.